

# Implications of altered sirtuins in metabolic regulation and oral cancer

Xu Quan[1], Ying Xin[2,3], He-Ling Wang[4], Yingjie Sun[5], Chanchan Chen[6] and Jiangying Zhang[5]

[1] Department of Stomatology, Shanghai General Hospital, Shanghai, China
[2] Key Laboratory of Shaanxi Province for Craniofacial Precision Medicine Research, College of Stomatology, Xi'an Jiaotong University, Xi'an, Shaanxi, China
[3] Department of Pathology, College of Stomatology, Xi'an Jiaotong University, Xi'an, Shaanxi, China
[4] Department of Clinical Molecular Biology, University of Oslo and Akershus University Hospital, Lørenskog, Norway
[5] Xiangya School of Stomatology, Central South University, Changsha, Hunan, China
[6] Department of Stomatology, Shenzhen Children's Hospital, Shenzhen, Guangdong, China

## ABSTRACT

Sirtuins (SIRTs 1-7) are a group of histone deacetylase enzymes with a wide range of enzyme activities that target a range of cellular proteins in the nucleus, cytoplasm, and mitochondria for posttranslational modifications by acetylation (SIRT1, 2, 3, and 5) or ADP ribosylation (SIRT4, 6, and 7). A variety of cellular functions, including mitochondrial functions and functions in energy homeostasis, metabolism, cancer, longevity and ageing, are regulated by sirtuins. Compromised sirtuin functions and/or alterations in the expression levels of sirtuins may lead to several pathological conditions and contribute significantly to alterations in metabolic phenotypes as well as oral carcinogenesis. Here, we describe the basic characteristics of seven mammalian sirtuins. This review also emphasizes the key molecular mechanisms of sirtuins in metabolic regulation and discusses the possible relationships of sirtuins with oral cancers. This review will provide novel insight into new therapeutic approaches targeting sirtuins that may potentially lead to effective strategies for combating oral malignancies.

## INTRODUCTION

Sirtuins (SIRTs) are nicotinamide dinucleotide (NAD$^+$)-dependent histone deacetylases and/or ADP-ribosyltransferases that have regulatory functions in a wide range of pathways involved in health and disease (*Chalkiadaki & Guarente, 2015*; *Imai et al., 2000*). A total of seven sirtuins, namely, silent information regulator 1 (SIRT1) to SIRT7, have been found in mammals and have been associated with the regulation of an impressive range of cellular processes, including DNA repair, cell survival and senescence, inflammation, metabolism, tumorigenesis, and healthy longevity (*Chalkiadaki & Guarente, 2015*). The oral cavity is the most common site of cancer in the head and neck region, and oral squamous cell carcinoma (OSCC) accounts for more than 90% of all oral malignancies. OSCC shows poor prognosis and high mortality. The molecular pathogenesis of OSCC is complex, resulting from a wide range of events that involve metabolites (*Vitório et al., 2020*). Sirtuins regulate numerous

Corresponding author
Jiangying Zhang,
zhjianying@csu.edu.cn

processes in OSCC, including tumour oncogenesis, metastasis, and chemoresistance (*Ezhilarasan et al., 2022*; *Chen et al., 2014*; *Xiong et al., 2011*). In this review, we provide an overview and an update on sirtuin functions in metabolism and with a specific focus on their role in oral cancer. A better understanding of SIRT biology at both the molecular and physiological levels will be essential for the future development of new treatments for oral cancer and it would be of particular interest to clinicians and researchers in the field of stomatology.

## Survey Methodology

This review describes sirtuins functions and the possible relationships of sirtuins with oral cancers. The keywords used in this review included sirtuin, metabolic regulation, and oral cancer, and all academic articles up to 2022 in relevant topics were searched through Google Scholar, Web of Science, and PubMed Central platform. Figures were generated using the subscription software BioRender (Toronto, ON, Canada).

# THE BASICS CHARACTERISTICS OF SEVEN MAMMALIAN SIRTUINS

## Classifications, structures, and subcellular locations

In mammals, the sirtuin family comprises seven proteins denoted as SIRT1-SIRT7. A phylogenetic analysis of 60 core domains found in different eukaryotes and prokaryotes revealed that four classes of sirtuins are found in mammals (I–IV) (*Teixeira et al., 2020*). SIRT1, SIRT2, and SIRT3 are members of the class I family of sirtuins, which is further subdivided into a, b, and c. SIRT1 belongs to Class I-a, which also includes *Saccharomyces cerevisiae* Sirt2 and Hst1, *Caenorhabditis elegans* Sirt−2.1, and *Drosophila melanogaster* D.mel1. SIRT2 and SIRT3 belong to Class I-b, which also includes yeast Hst2, fly D.mel2, and other sirtuins found in some bacteria and fungi. SIRT4 is a member of Class II, which also includes sirtuins from bacteria, insects, nematodes, mould fungi, and protozoa. Sirtuin 5 is a mammalian member of the Class III sirtuins, which are widely distributed among all eukaryotes, including bacteria and archaea. Class IV includes SIRT6 and SIRT7 in two different subclasses, IV-a and IV-b (*Mostoslavsky et al., 2006*), which are widespread in metazoans, plants, and vertebrates (*Jiao & Gong, 2020*; *Schuetz et al., 2007*).

The seven mammalian SIRTs, all of which are widely distributed in cells, share a highly conserved catalytic core domain flanked by distinct NH2- and COOH-terminal regions (*Haigis & Sinclair, 2010*) (Fig. 1), as demonstrated by primary sequence alignments. Two domains are conserved in the sirtuin enzymatic core: a large Rossmann fold domain that binds $NAD^+$ and a small domain formed by insertions of the large domain that binds to zinc atoms. The diversity of amino acid sequences in distinct N- and/or C-terminal extensions of different sirtuins accounts for their different subcellular localizations, substrate binding abilities, catalytic activities, and physiological functions (*Haigis & Sinclair, 2010*).

A high level of fidelity has been observed in the catalytic/enzymatic cores of the seven sirtuins. The first sirtuin structure was identified in 2001 by *Finnin, Donigian & Pavletich (2001)*. SIRT2 was the first reported subtype and nearly represents the structural basis for all sirtuins in their enzymatic cores. The catalytic core consists of two main parts: a

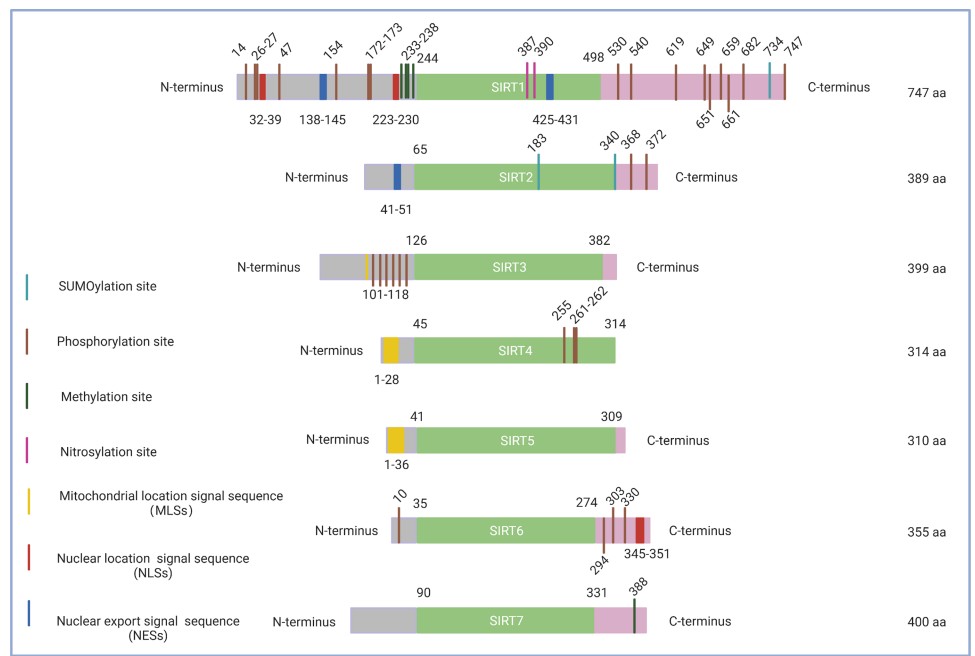

**Figure 1    Schematic structure of the sirtuin family.** Only the canonical isoform of each sirtuin is shown. The green boxes indicate the core domains of each sirtuin. (1) SIRT1 consists of 747 amino acids with 16 phosphorylation sites, four methylation sites, two nitrosylation sites, a SUMOylation site, two nuclear location signal sequences (NLSs), and two nuclear export signal sequences (NESs). (2) SIRT2 exists as a long chain of 389 amino acid molecules, which includes two phosphorylation sites, two SUMOylation sites, and one nuclear export signal sequence (NES). (3) SIRT3 contains six phosphorylation sites located between the amino acids 101 to 118 and a mitochondrial location signal sequence (MLS) in the C-terminal extension. (4) SIRT4, which consists of 314 amino acids, possesses a mitochondrial location signal sequence (MLS) in the N-terminal region and three phosphorylation sites at Ser255, Ser261, and Ser262. (5) SIRT5 features 310 amino acids and a mitochondrial location signal sequence (MLS) consisting of 36amino acids. (6) SIRT6 has three C-terminal phosphorylation sites and one N-terminal phosphorylation site as well as a nuclear location signal sequence (NLS) between amino acids 345 and 351. (7) SIRT7 has a methylation site located at Arg388 in its C-terminal region. Figure created by BioRender (Toronto, ON, Canada).

conserved large Rossmann fold domain and a variable small domain (*Finnin, Donigian & Pavletich, 2001*). The large Rossmann fold domain is inverted and consists of 6 $\beta$-strands and 6$\alpha$-helices, and the small domain contains a helical module and a $Zn^{2+}$ finger module (*Bellamacina, 1996*). Both modules of the small domain are connected to the large Rossmann fold domain to form a large groove between the two domains. In addition, variable specialized domains, including nuclear localization signal (NLS) sequences, nuclear export signal (NES) sequences, and mitochondrial targeting sequences (MTS), control the subcellular localizations and distributions of sirtuins, which are crucial for their function (*Sanders, Jackson & Marmorstein, 2010*) (Fig. 1). It is worth noting that although sirtuins may have a similar biochemical function in some cases, they could play different biological roles determined by their intracellular compartmentalization and their expression patterns within tissues.
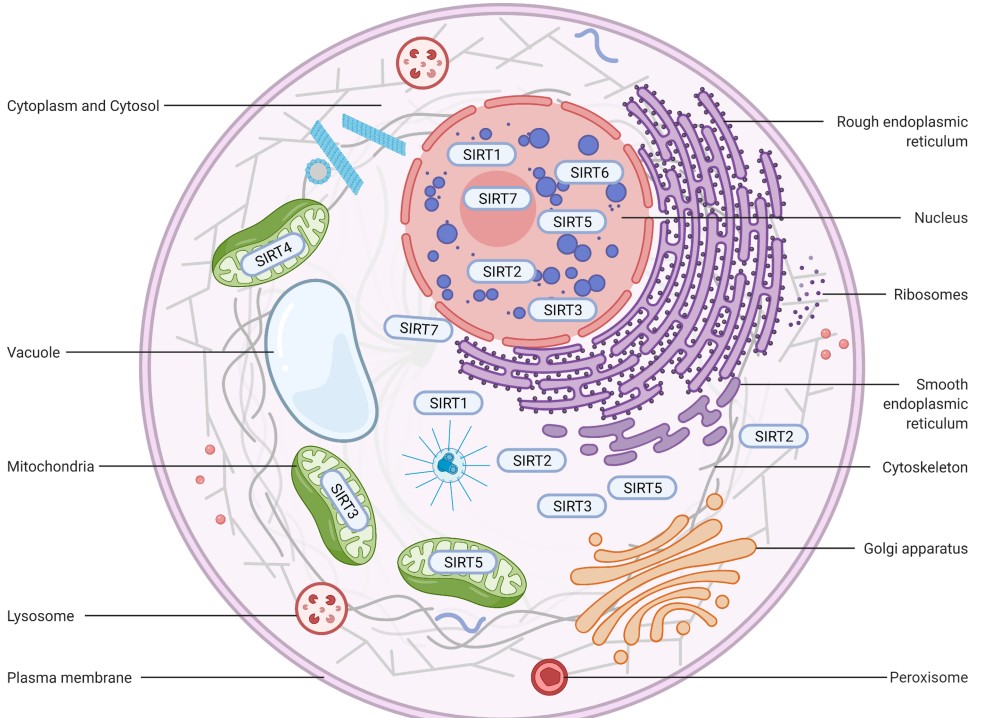

**Figure 2** **Subcellular location of the sirtuin family.** (1) SIRT1 is mainly located in the nucleus and cytoplasm. (2) SIRT2 can proactively shuttle between the nucleus and cytoplasm and is primarily found in the cytoplasm, cytosol, and cytoskeleton. (3) SIRT3, SIRT4, and SIRT5 predominantly reside in mitochondria. SIRT3 and SIRT5 can be found extra-mitochondrially. (4) SIRT6 is chiefly a nuclear protein. (5) SIRT7 predominantly resides in the nucleolus and nucleoplasm and is observed near the nuclear membrane in the cytoplasm and cytosol. Figure created by BioRender (Toronto, ON, Canada).

### SIRT1

SIRT1 is the most extensively investigated of the seven mammalian sirtuins (*Brooks & Gu, 2009*). SIRT1 exhibits the highest homology with yeast Sir2, which delays the ageing process and prolongs the lifespan in Saccharomyces cerevisiae, Caenorhabditis elegans, and Drosophila melanogaster under caloric restriction (CR) (*Burnett et al., 2011*; *Kaeberlein, McVey & Guarente, 1999*; *Chen & Guarente, 2007*). The SIRT1 protein is largely found in the nucleus but can also be shuttled between the cytosol and nucleoplasm in various tissues in response to different environmental signals (*Tanno et al., 2007*) (Fig. 2). The SIRT1 protein contains 747 amino acids and has three independent domains: a central deacetylase domain (244–512 residues) that is conserved among species, a nuclear localization/export signal domain located in the N-terminal region (513–747 residues), and an essential activity domain located in the C-terminal region (1–180 residues). Specifically, the catalytic domain houses a substrate and an NAD$^+$-binding pocket, and the N- and C-termini contain the regulatory and binding domains of the SIRT1 coactivator/corepressor, respectively. Additionally, SIRT1 contains an NLS (KRKKRK) at residues 41–46 and has thus been labelled a nuclear protein (*Frye, 1999*).

### SIRT2

SIRT2 is the mammalian orthologue of yeast Hst2 (*Perrod et al., 2001*). Similar to yeast Sir2, Hst2 is upregulated by CR and oxidative stress and extends the lifespan through a Sir2-independent pathway (*Lamming et al., 2005*). SIRT2 is predominantly found in the cytoplasm (Fig. 2), where it acts principally as a deacetylase of microtubular proteins, such as alpha-tubulin (*North et al., 2003*). Thus, it serves as a regulator of the cell cycle, division, and proliferation (*Li et al., 2007*). Several lines of evidence suggest that SIRT2 has the ability to shuttle between the cytoplasm and the nucleus *via* cis-regulatory module (CRM)1-dependent active nuclear export (*Inoue et al., 2007*). The localization and function of SIRT2 are dependent on the differential splicing of its RNA (RNA splicing), which produces distinct isoforms with different N- and C-terminal extensions. Four different slice variants (also known as isoforms) are currently reported in the GenBank sequence database. However, only isoforms 1 and 2 have confirmed protein products with biological functionality. Isoform 1 is the full protein (389 aa), and isoform 2 lacks the first 37 aa from the N-terminus. Both isoforms 1 and 2 have a highly conserved catalytic core domain consisting of approximately 276 amino acid residues. A leucine-rich NES within the N-terminal region of these two isoforms has also been characterized (*Pereira et al., 2018*). Deletion of the NES leads to nucleocytoplasmic distribution, which suggests that the NES mediates cytosolic localization (*North & Verdin, 2007*).

### SIRT3

SIRT3, which is a major $NAD^+$-dependent protein deacetylase in mitochondria, plays an important role in regulating mitochondrial metabolism and energy production and is thought to be responsible for both the positive effects of exercise and caloric restriction on health (*Schwer et al., 2002*). Most studies support the notion that SIRT3 is localized in mitochondria (*Onyango et al., 2002*; *Lombard et al., 2007*; *Verdin et al., 2010*; *Kratz et al., 2021*), whereas other studies have suggested that SIRT3 might also be localized in the nucleus and cytoplasm (*Scher, Vaquero & Reinberg, 2007*; *Sundaresan et al., 2008*) (Fig. 2). Therefore, the localization and function of SIRT3 in various cellular compartments remain controversial. As a typical sirtuin, SIRT3 has a conserved enzymatic core (126–382 aa) responsible for its deacetylation function and acts in an $NAD^+$-dependent manner. Two isoforms of SIRT3 are produced by alternative splicing in human cells.

The full-length 44-kDa form (isoform 1) contains 399 amino acid residues and is cleaved within mitochondria by matrix metalloprotease to a 28-kDa short form (which is denoted as isoform 2), which consists of an N-terminus missing 1–142 amino acid residues (*Ota et al., 2004*). The long isoform is found in mitochondria, the cytoplasm, and the nucleus, whereas the short isoform is found only in mitochondria (*Scher, Vaquero & Reinberg, 2007*).

### SIRT4

In contrast to other sirtuins, SIRT4 has been relatively less extensively investigated. However, SIRT4 shares a conserved catalytic core of −270 amino acids. The core of SIRT4 has no C-terminal domain and contains a short, ~44-aa N-terminal extension, which serves as a mitochondrial localization signal sequence (MLS) (*Verdin et al., 2010*; *Kratz*

*et al., 2021*). The catalytic part of SIRT4 has a typical structure, which contains a large Rossmann fold domain and a small domain. The presence of an N-terminal MLS ensures that SIRT4 localizes within the mitochondrial matrix (Fig. 2). The cleavage of SIRT4 at amino acid 28 after its import into the mitochondria activates the enzymatic functions of the protein (*Ahuja et al., 2007*).

### SIRT5

SIRT5, which is another mitochondrial sirtuin (mtSIRT), is the most recently investigated sirtuin. Similar to SIRT3 and SIRT4, SIRT5 is predominantly located in the mitochondrial matrix due to the presence of an N-terminal MTS (*Verdin et al., 2010*; *Kratz et al., 2021*). However, several studies have reported that SIRT5 is also found outside the mitochondria, although a fraction is observed in the cytosol (*Matsushita et al., 2011*) and peroxisomes (*Chen et al., 2018*), and very small amounts are also detected in the nucleus (*Park et al., 2013*) (Fig. 2). In humans, the SIRT5 gene encodes two major isoforms of the protein. SIRT5$^{iso1}$ is composed of 310 amino acids, whereas SIRT5$^{iso2}$ contains 299 amino acids and has a C-terminus that differs slightly from that of SIRT5$^{iso1}$ (*Matsushita et al., 2011*). More specifically, SIRT5$^{iso2}$ has 14 different residues (SHLISISSLIIIKN) between residues 286 and 299 and a missing aa in the 310th residue. Two additional human SIRT5 isoforms (SIRT5$^{iso3}$ and SIRT5$^{iso4}$) are also in the NCBI database (*NR, 2018*). The sequence of SIRT5$^{iso3}$ resembles the sequence of SIRT5$^{iso1}$ with the exception of a lack of 18 internal amino acids (aa 189–206 are not present). SIRT5$^{iso4}$ is missing the first 108 amino acids of SIRT5$^{iso1}$, includes the MTS, and completely aligns with amino acids 109–310 of SIRT5$^{iso1}$. No data are currently available regarding the expression, localization, or functional properties of SIRT5$^{iso3}$ and SIRT5$^{iso4}$. A comparison of the structures of SIRT5 with those of other sirtuins reveals that the overall domain organization and folding of SIRT5 are similar to those of other reported sirtuin structures and that SIRT4 and SIRT5 exhibit many more overlapping regions (*Soding, 2005*).

### SIRT6

SIRT6, a member of the sirtuin family of NAD$^+$-dependent deacetylases, plays an important role in biological homeostasis, longevity, and various disease conditions. The SIRT6 structure consists of a large Rossmann fold and a smaller and structurally more varied sequence containing a zinc-binding domain. Unlike other sirtuins, SIRT6 does not contain a highly conserved cofactor-binding loop that aids in NAD$^+$ binding but rather exhibits a helix structure that forms interactions with both ADP-ribose and 2′-N-acetyl-ADP-ribose (NAADPr) (*Pan et al., 2011*). The C-terminus is needed for proper nuclear localization, whereas the N-terminus is involved in the formation of chromatin associations and in enzymatic activity (*Tennen, Berber & Chua, 2010*). SIRT6 is reportedly a predominantly nuclear protein (*Mostoslavsky et al., 2006*) associated with telomeric heterochromatin regions (*Michishita et al., 2008*) (Fig. 2). The full-length isoform of SIRT6 (isoform 1, 39.1 kDa) contains 355 amino acids, and the shorter isoform (isoform 2, 36 kDa) lacks amino acids from the catalytic domain (amino acids 179–205) (*Miteva & Cristea, 2014*).

### SIRT7

In humans, *SIRT7* encodes a 400-amino-acid protein (in its full-length form) that functions as a class IV histone deacetylase that plays diverse roles in the ageing process, metabolic stress, and disease biology. Similar to other nuclear-localized sirtuins flanking the NLS (SIRT1 and SIRT6), SIRT7 is highly enriched in the nucleolus (*Ford et al., 2006*). A proportion of SIRT7 is also found close to the nuclear membrane in the cytoplasm (*Kiran et al., 2013*), which suggests that the shuttling of SIRT7 between various subcellular compartments is associated with and may be responsible for its multiple effects in diverse cellular responses (*Tang et al., 2019*; *Zhang et al., 2016*) (Fig. 2). Three protein-coding isoforms of SIRT7 are identified in the UniProtKB database due to alternative splicing mechanisms: (a) Q9NRC8-1, isoform 1, 400 aa, 44.9 kDa; (b) Q9NRC8-2, isoform 2, 183 aa, 20.4 kDa; and (c) Q9NRC8-3, isoform 3, 320 aa, 35.9 kDa. A search of the Ensembl Genome Browser revealed 21 splice variants in transcription products (ENSG00000187531), but there were only two protein-coding variants (SIRT7-210 and SIRT7-201 isoforms) (https://www.ensembl.org/index.html). To date, only a fragment of the SIRT7 N-terminus has been experimentally resolved, and the structure of the whole molecule remains to be determined (*Priyanka et al., 2016*). Based on a phylogenetic analysis, SIRT7 exhibits the highest degree of similarity to SIRT6 (*Costantini et al., 2013*).

## General catalytic activities

A study in 2,000 provided the first demonstration that Sir2 has robust histone deacetylase activity that requires nicotinamide adenine dinucleotide ($NAD^+$) as an obligate cosubstrate (*Imai et al., 2000*). Sirtuins can sense the level of $NAD^+$ in cells to catalyse protein lysine deacetylation by modulating the properties and functions of proteins, such as histones, kinases, and transcription factors (TFs), by removing acetyl groups posttranslationally attached to their lysine residues. Deacetylation reactions consume one molecule of $NAD^+$ and produce 2′-O-acetyl ADP-ribose and nicotinamide (NAM) (*Sauve et al., 2001*). Given the crucial roles of $NAD^+$ in energy production, health, and longevity, researchers are motivated to explore the notion of supplementing $NAD^+$ biosynthesis precursors to increase health benefits (*Gilmour et al., 2020*).

As mentioned above, sirtuins are class III histone deacetylases (HDACs) whose activities are dependent on $NAD^+$ levels and thus on the metabolic status of cellular organelles. Due to this feature, the activity of sirtuins is coupled to the cellular metabolic status (*Covington & Bajpeyi, 2016*), which allows enzymes to modulate the proteins of the electron transport chain (ETC), the stress response, and live-and-death signalling. In addition to their primary functions, sirtuins also have additional enzymatic activities, such as mono (ADP-ribosylation) activity (SIRT3, 4 and 6), the ability to remove a wide range of other lysine modifications (*e.g.*, desuccinylation and demalonylation with SIRT5 and decrotonylation with SIRT1, 2, and 3), and the absence of deacetylation capabilities (SIRT4) (*Kupis et al., 2016*; *Jesko et al., 2017*; *Jesko & Strosznajder, 2016*). It has become increasingly clear that sirtuins are involved in a variety of interdependent processes, including crosstalk with transcription factors, such as forkhead box subgroup O (FOXO), p53, NF-$\kappa$B, and proteins involved in DNA damage repair (*Avilkina, Chauveau & Ghali Mhenni, 2022*). It is striking
to note that the versatile and abundant macromolecules poly (ADP-ribose) polymerases (PARPs) bear the same characteristics as sirtuins in that they share a dependence on NAD$^+$ for their substrate conversion and exhibit a variety of interactions, which influence a wide range of functions in cells (*Kupis et al., 2016*; *Jesko et al., 2017*; *Jesko & Strosznajder, 2016*).

# SIRTUINS IN METABOLIC REGULATION

A mounting body of evidence has shed light on the fact that sirtuins play diverse roles during the course of metabolism. There is a constant balance between the flow of molecules through metabolic pathways and the utilization of energy by cells. Here, the metabolic capacities of sirtuins, with emphasis on how they regulate glucose, lipid, and protein metabolism, are discussed in detail (Table 1).

## Glucose metabolism

Metabolic processes involving glucose include glucose uptake, utilization, storage, and output, which require extensive cooperation between insulin and its regulating hormone counterpart, glucagon. In addition to their function as transcription factors, sirtuins have received considerable attention regarding their role in regulating and maintaining gluconeogenesis, glycolysis, and insulin secretion.

### SIRT1

SIRT1 is of central importance in regulating gluconeogenesis through its ability to deacetylate target proteins. SIRT1 can deacetylate CREB-regulated transcription coactivator 2 (CRTC2), which causes CRTC2 degradation and decreases hepatic glucose production (*Liu et al., 2008*). SIRT1 also enhances hepatic glucose output and gluconeogenesis through peroxisome proliferator-activated receptor (PPAR)g coactivator 1a (PGC-1$\alpha$) and forkhead box O1 (FOXO1) (*Houtkooper, Pirinen & Auwerx, 2012*; *Rodgers et al., 2005*). PGC-1$\alpha$ is an important substrate of SIRT1 that plays a vital role in modulating glucose metabolism. Through PGC-1$\alpha$, SIRT1 induces gluconeogenic genes in the liver. In contrast, in response to fasting and pyruvate, SIRT1 can modulate the PGC-1$\alpha$-induced repression of glycolytic genes (*Rodgers et al., 2005*). Regarding glycolysis, SIRT1 can inhibit the process of glycolysis through the deacetylation and repression of glycolytic enzymes, such as phosphoglycerate mutase-1 (PGAM-1) (*Hallows, Yu & Denu, 2012*). SIRT1 reportedly suppresses glycolysis by repressing hypoxia-inducible factor 1$\alpha$ (HIF-1$\alpha$) (*Houtkooper, Pirinen & Auwerx, 2012*; *Lim et al., 2010*). In pancreatic $\beta$ cells, SIRT1 regulates insulin secretion by inhibiting the expression of UCP-2 and increasing ATP production to shut down the potassium channel, which allows the entry of calcium and the release of insulin (*Bordone et al., 2006*). In addition, SIRT1 promotes insulin expression by activating the expression of NeuroD and MafA (*Zhou, Tang & Chen, 2018*) SIRT1 and its activators reduce insulin resistance and diabetic complications and are thus potentially effective therapeutic targets for type 2 diabetes (T2D) (*Ma et al., 2016*; *Kitada et al., 2019*; *Hegedus et al., 2020*).

### SIRT2

In terms of biochemical activities, SIRT2 is most similar to SIRT1, and its deacetylase activity can also promote glyconeogenesis. For example, SIRT2 can stabilize and

**Table 1  Sirtuins related metabolism.**

| Sirtuins | Regulatory factors | Functions |
|---|---|---|
| SIRT1 | CRTC2↓ | Gluconeogenesis↓ |
| | PGC-1α↑ FOXO1↑ | Gluconeogenesis↑ |
| | PGC-1α↑ PGAM-1↓HIF-1α↓ | Glycolysis↓ |
| | UCP-2↓ NeuroD and MafA↑ | Insulin↑ |
| | (SREBP)-1 and (SREBP)-2↓ | Lipid synthesis↓ |
| | PPARγ↓ | Fat mobilize on↓ |
| | PGC-1α and PPARα↑ | Fatty acid use↓ |
| SIRT2 | PEPCK↓ | Gluconeogenesis↑ |
| | FOXO1 and PPARγ↓ HNF4α↑ | Adipogenesis↓ |
| SIRT3 | HIF-1α↓ HK2↓ | Glycolysis↓ |
| | Khib and PFK↑ | Glycolysis↑ |
| | GDH↑ | Glucose synthesis↑ |
| | LCAD↓ AMPK↑ | Fatty acid oxidation↑ |
| | SCD1↓ | Lipogenesis↓ |
| | HMGCS2↑ | Ketogenesis↑ |
| SIRT4 | GDH↓ | Insulin↑ |
| | Leucine catabolism↑ | Insulin↓ |
| | MCD↓SIRT1 and PPARα↓ | Fatty acid oxidation↓ |
| | PPARγ↑ | Adipogenesis↑ |
| SIRT5 | GAPDH↑ | Glycolysis↑ |
| | PPARγ and Prdm16↑ | Brown adipogenesis↓ |
| SIRT6 | HIF-1α↓ | Glycolysis↓ |
| | 5GCN5↑ | Gluconeogenesis↓ |
| | AMPKα↑ PPARγ↓ | Lipid synthesis↓ |
| SIRT7 | HIF-1α↓ HIF-2α↓ | Glycolysis↓ |
| | SIRT1↓ PPARγ↑ | Adipogenesis↑ |

**Notes.**
↓ represents that the targets are inhibited or repressed by Sirtuins.
↑ represents those are activated or promoted by Sirtuins.

deubiquitinate phosphoenolpyruvate carboxykinase (PEPCK-C), a rate-limiting enzyme in gluconeogenesis. During glucose deprivation, SIRT2 deacetylates PEPCK and increases gluconeogenesis (*Jiang et al., 2011*). SIRT2 maintains insulin sensitivity by acting as a glucose sensor. SIRT2 plays a vital role in supporting insulin resistance, and downregulation of SIRT2 improves insulin sensitivity (*Lemos et al., 2017*). A recent study also showed that SIRT2 ablation impairs glucose-stimulated insulin secretion by blocking glucokinase regulatory protein degradation and promoting aldolase A protein degradation, which causes a reduction in glycolytic flux (*Zhou et al., 2021*).

### SIRT3

In addition to SIRT1, SIRT3 reportedly regulates glycolytic metabolism by maintaining the stability and regulating the activity of HIF-1α (*Finley et al., 2011*; *Wang et al., 2020*; *Katwal et al., 2018*). In contrast, a reduced level of SIRT3 is associated with high acetylation of peptidylprolyl isomerase D (cyclophilin D), which activates hexokinase II (HK2), a critical

enzyme in glycolysis pathways (*Wei et al., 2013*). According to a recent report, the absence of SIRT3 is related to increases in the lysine 2-hydroxyisobutyrylation (Khib) levels of phosphofructokinase (PFK) and in glycolysis (*Perico et al., 2021*). SIRT3 can also initiate glucose synthesis by activating glutamate dehydrogenase (GDH) (*Li et al., 2019*; *Fu et al., 2022*), which converts glutamate to $\alpha$-ketoglutarate in mitochondria (*Schlicker et al., 2008*; *Zou et al., 2017*).

### SIRT4

Unlike SIRT1-3, SIRT4 does not display NAD-dependent deacetylase activity and can regulate insulin secretion by using NAD for the ADP-ribosylation of GDH in pancreatic $\beta$ cells (*Haigis et al., 2006*). SIRT4 downregulates the enzymatic activity of GDH and hinders the production of ATP from glutamate and glutamine to further promote insulin secretion (*Haigis et al., 2006*). In addition to GDH, SIRT4 is thought to regulate insulin secretion *via* various targets, including ADP/ATP carriers and the insulin-degrading enzymes ANT2 and ANT3 (*Ahuja et al., 2007*). Furthermore, SIRT4 can inhibit insulin secretion by promoting leucine catabolism (*Wang & Wei, 2020*).

### SIRT5

SIRT5 possesses deacetylase- and $NAD^+$-dependent demalonylase and desuccinylase activities. Glyceraldehyde-3-phosphate dehydrogenase (GAPDH) is a glycolytic enzyme. In glycolysis, SIRT5 can regulate the activity of GAPDH by demalonylating its homodimerization interface residue, K184 (*Nishida et al., 2015*). The findings of a recent study suggest that SIRT5 may be positively correlated with insulin sensitivity (*Jukarainen et al., 2016*). Although SIRT5 plays multiple roles in the regulation of cellular metabolism, further research is needed to identify its direct substrates and determine its exact function.

### SIRT6

SIRT6 is essential for the maintenance of glucose homeostasis. Similar to SIRT1, SIRT6 suppresses glycolysis by acting as a corepressor for HIF-1$\alpha$ (*Zhong et al., 2010*; *Zeng et al., 2021*). In glyconeogenesis, SIRT6 binds to and promotes the activity of 5GCN5 (general control nonrepressed protein), which acetylates PGC-1$\alpha$. Acetylated PGC-1$\alpha$ activates PPAR $\gamma$ to inhibit glyconeogenesis-related enzymes, such as PEPCK-C and G6P, thereby resulting in the inhibition of hepatic glucose production by repressing gluconeogenesis (*Kugel & Mostoslavsky, 2014*). SIRT6 also maintains glucose homeostasis by downregulating multiple members of the insulin signalling pathway, such as AKT, insulin receptor, and the insulin receptor substrates IRS1 IRS2, glucose transporter-1 (GLUT1), and glucose transporter-4 (GLUT4) (*Xiao et al., 2010*; *Parenti et al., 2014*; *Liu et al., 2018*; *Huang et al., 2019*; *Yang et al., 2020*; *Wu et al., 2021*; *Tang & Fan, 2019*).

### SIRT7

SIRT7 also interacts with hypoxia-inducible factors. The overexpression of SIRT7 can reduce the protein levels of both HIF-1$\alpha$ and HIF-2$\alpha$ independently of its deacetylase activity (*Hubbi et al., 2013*; *Wu et al., 2018*). Moreover, mice lacking SIRT7 display better resistance to glucose intolerance and increased insulin sensitivity when fed fat-containing

diets, which suggests that SIRT7 plays a crucial role in glucose metabolism (*Yoshizawa et al., 2014*).

## Lipid metabolism

Lipid metabolism includes lipid synthesis and lipolysis. By controlling lipid metabolism, cells and tissues can obtain lipid materials and meet their energy needs. The up- or downregulation of specific transcription factors, which can alter the rate of lipid synthesis or lipolysis by targeting specific genes, is one of the most effective ways to regulate lipid homeostasis. Sirtuins can regulate lipid metabolism by interacting with some vital transcription factors.

### SIRT1

SIRT1 can regulate lipid metabolism *via* its deacetylase activity. For instance, SIRT1 deacetylates and destabilizes sterol regulatory element-binding protein (SREBP)-1 and (SREBP)-2, which are transcription factors related to lipid metabolism, and thereby represses lipid synthesis and fat storage during fasting (*Thiel, Guethlein & Rossler, 2021*). In white adipose tissue, SIRT1 mediates corresponding effects on fat accumulation. SIRT1 binds to and functionally inhibits the fat regulator peroxisome proliferator-activated receptor-$\gamma$ (PPAR$\gamma$) by interacting with the PPAR$\gamma$ cofactor nuclear receptor corepressor (NCoR) and silencing the mediator of retinoid and thyroid hormone receptors (SMRT) (*Zhou, Tang & Chen, 2018*; *Picard et al., 2004*). A SIRT1/PPAR$\gamma$/NCoR complex binds to conspecific DNA sites in PPAR-$\gamma$ target gene promoter sequences and suppresses their transcription (*Picard et al., 2004*). Thus, genes involved in fatty acid accumulation and lipolysis can thus be negatively affected. SIRT1 also regulates hepatic lipid homeostasis by interacting with peroxisome proliferator-activated receptor -$\alpha$ (PPAR$\alpha$), a nuclear receptor for lipid homeostasis (*Bougarne et al., 2018*). PGC-1$\alpha$ is a direct substrate of PPAR$\alpha$ (*Vega, Huss & Kelly, 2000*; *Cheng, Ku & Lin, 2018*), and SIRT1 alters PPAR$\alpha$ signalling by deacetylating and activating the PPAR$\alpha$ coactivator PGC-1$\alpha$ (*Purushotham et al., 2009*; *Kalliora et al., 2019*; *Kosgei et al., 2020*; *Li et al., 2021*). The loss of SIRT1 reduces PPAR$\alpha$ signalling and impairs fatty acid $\beta$-oxidation (*Purushotham et al., 2009*). A growing body of evidence suggests that SIRT1 could be an important therapeutic target in preventing lipid metabolic diseases.

### SIRT2

SIRT2 exerts a negative regulatory effect on adipogenesis through its deacetylase activity. By deacetylating FOXO1, SIRT2 suppresses adipogenesis in part through binding FOXO1 to PPAR$\gamma$ and repressing its transcriptional activity (*Wang & Tong, 2009*). A recent study revealed that SIRT2 inhibits lipid accumulation partially by binding to and deacetylating the hepatocyte nuclear factor 4$\alpha$ (HNF4$\alpha$) protein on lysine 458 to increase HNF4$\alpha$ stability (*Ren et al., 2021*). SIRT2 may be a promising target in the treatment of lipid metabolic disorders.

### SIRT3

SIRT3 plays an essential role in the metabolic process of fatty acid oxidation (FAO). SIRT3 can deacetylate and reduce the enzymatic activity of long-chain acylCoA dehydrogenase

(LCAD), a protein involved in FAO, during prolonged fasting to enhance FAO (*Hirschey et al., 2010*). According to a recent study, SIRT3 also regulates FAO by deacetylating liver kinase B1 (LKB1) and activating AMP-activated protein kinase (AMPK) (*Li et al., 2020*). Additionally, SIRT3 contributes to the prevention of nonalcoholic fatty liver disease. SIRT3 ameliorates lipotoxicity in hepatocytes by reducing the expression of stearoyl-CoA desaturase 1 (SCD1), a key lipogenic enzyme, to suppress lipogenesis (*Zhang et al., 2020*). SIRT3 can also deacetylate and stimulate the activity of 3-hydroxy-3-methylglutaryl CoA synthase 2 (HMGCS2) in the liver (*Hirschey et al., 2011*), which results in increased ketogenesis (*Shimazu et al., 2010*).

### SIRT4

In contrast to SIRT3, SIRT4 negatively regulates FAO and stimulates lipogenesis by directly binding to, deacetylating, and repressing malonyl-CoA decarboxylase (MCD), an enzyme that produces acetyl-CoA from malonyl-CoA (*Laurent et al., 2013b*). In addition, by dampening the activity of SIRT1 and PPAR$\alpha$, SIRT4 can lead to a reduction in FAO in the liver (*Laurent et al., 2013a*). A recent study showed that SIRT4 positively functions as a regulator of branched-chain amino acid (BCAA) catabolism, promotes the expression of PPAR$\gamma$ in early adipogenesis, and consequently stimulates adipogenesis (*Zaganjor et al., 2021*).

### SIRT5

The hepatic overexpression of SIRT5 can improve mitochondrial FAO in hepatocytes by desuccinylation proteins (*Du et al., 2018*). SIRT5-knockout mice also exhibit reduced FAO (*Rardin et al., 2013*). In parallel, SIRT5 can protect against acute kidney injury by regulating proximal tubule FAO (*Chiba et al., 2019*). The effect of SIRT5 on FAO may have potential therapeutic implications in the treatment of acute kidney injury. In addition, SIRT5 deficiency reduces the intracellular levels of $\alpha$-ketoglutarate, and this reduction leads to higher levels of methylation at the promoters of the PPAR$\gamma$ and Prdm16 genes, which can repress brown adipogenesis (*Seale, Kajimura & Spiegelman, 2009*). According to a recent study, a SIRT5 inhibitor stimulates brown adipogenesis (*Molinari et al., 2021*).

### SIRT6

SIRT6 plays a crucial role in lipid mobilization. By activating the adenosine monophosphate-activated protein kinase alpha (AMPK $\alpha$) pathway, SIRT6 inhibits preadipocyte differentiation and lipid synthesis and works in concert with SIRT5 to decrease lipid deposition and inhibit cell cycle arrest of preadipocytes (*Hong et al., 2020*). Additionally, SIRT6 deficiency causes increases in triglyceride (TG) synthesis and long-chain fatty acid uptake and decreases fatty acid $\beta$-oxidation genes. Furthermore, the knockout of SIRT6 results in fatty acid liver disease due to TG accumulation (*Kim et al., 2010*). Under high-fat diets, SIRT6 maintains lipid homeostasis by downregulating genes specifically regulated by PPAR$\gamma$ (*Kanfi et al., 2010*). Furthermore, SIRT6 can bind to the DNA-binding domain of PPAR$\gamma$, and this binding regulates its activity at promoters and consequently controls the expression of fatty acid transporters (*Khan et al., 2021*).

### *SIRT7*

SIRT7 can interact with SIRT1 during adipogenesis. By inhibiting the activity of SIRT1, SIRT7 contributes to efficient adipocyte differentiation and thereby indirectly and efficiently activates PPAR$\gamma$ (*Fang et al., 2017*). Furthermore, SIRT7 can bind to and directly deacetylate PPAR$\gamma$2 to regulate adipocyte lipogenesis (*Akter et al., 2021*). However, according to a previous study, SIRT7-knockout mice show liver steatosis as a result of suppressed ER stress (*Shin et al., 2013*). In light of this finding, further research on the mechanisms underlying the regulation of lipids by SIRT7 is needed.

## Protein metabolism

The role of sirtuins in protein synthesis is just beginning to be understood in terms of their function. SIRT1 positively regulates protein processing in the ER and controls the acetylation status of several proteins involved in ribosome biogenesis and rRNA processing and ribosomal proteins (*Gil et al., 2017*).

Mitochondrial ribosomes play a crucial role in protein synthesis. SIRT3 can regulate protein synthesis by deacetylating mitochondrial ribosomal protein L10 (MRPL10), which is the main acetylated protein in the mitochondrial ribosome that regulates mitochondrial protein synthesis (*Yang et al., 2010*). However, SIRT4 and SIRT5 do not affect the deacetylation of mitochondrial proteins (*Lombard et al., 2007*).

SIRT6 can regulate protein stability and function by its deacetylase activity. in the nucleus SIRT6 directly deacetylates Tau-K174ac, regulating its nuclear functions and leading to the global pattern of protein translation and synthesis (*Portillo et al., 2021*). In various cell types, SIRT6 negatively regulates protein synthesis independent of its deacetylase activity; for example, SIRT6 can control the expression of mTOR signalling and consequently regulate protein synthesis (*Ravi et al., 2019*).

SIRT7 can affect protein levels by regulating polymerase I (Pol I)-induced rDNA transcription (*Ford et al., 2006*). For instance, SIRT7 knockdown triggers the downregulation of protein levels in cells through the degradation of RNA Pol I transcription (*Tsai et al., 2012*). In contrast, by enhancing Pol I occupancy at rDNA genes, SIRT7 stimulates the transcription of rRNA genes, which confirms the interplay between SIRT7 and protein synthesis in an animal model (*Chen et al., 2013b*). In addition to its role in Pol I transcription, SIRT7 also regulates the transcription of snoRNAs and mRNAs *via* interaction with Pol II (*Blank et al., 2017*). Furthermore, SIRT7 knockdown suppresses protein synthesis and RNA transcription by regulating Pol III function through the recruitment of mTOR kinase to the vicinity of tRNA genes (*Tsai, Greco & Cristea, 2014*). Interestingly, SIRT7 knockdown preferentially suppresses protein synthesis rather than tRNA transcription (*Tsai, Greco & Cristea, 2014*).

## SIRTUINS IN ORAL CARCINOGENESIS

It is widely believed that sirtuins regulate numerous processes in cancer cells, such as tumour suppression/oncogenesis, epithelial–mesenchymal transition (EMT), cell cycle progression, and autophagy (*Ezhilarasan et al., 2022*). In this review, we focus on the

regulatory mechanisms of SIRTs and their potential molecular targets in oral cancer (summarized in Table 2) and discuss their importance as possible therapeutic targets.

## SIRT1

SIRT1 acts as a bifunctional factor in oral cancer (*Ezhilarasan et al., 2022*). On the one hand, SIRT1 works as a tumour suppressor. An *in vitro* analysis showed that SIRT1 overexpression inhibits the proliferation and invasiveness of human OSCC cell lines, such as SCC-9 and SCC-25 (*Kang et al., 2018a*). Clinical studies have shown that the SIRT1 level is significantly downregulated in patients with OSCC (*Chen et al., 2014*). EMT plays a key role in the regelation of cancer invasion and metastasis, during which epithelial cells lose their junction proteins, reduce epithelial cadherin (E-cadherin) and increase their levels of mobility (*Huang et al., 2022*; *Mishev et al., 2014*). Moreover, through increasing E-cadherin expression, SIRT1 is able to promote epithelial integrity in oral cancer cells, thereby suppressing invasion and metastasis (*Chen et al., 2014*). Additionally, SIRT1 suppresses mesenchymal makers N-cadherin and vimentin expression and downregulates migration and invasion genes, such as *csk2a2, fra1, actb, and slug*, preventing oral cancer (*Ezhilarasan et al., 2022*; *Murofushi et al., 2017a*). Transforming growth factor-beta (TGF-$\beta$) is an upstream signal regulating EMT and its expression can lead to malignant transformation, invasion and metastasis in oral epithelial cells by interacting with downstream targets (*Chen et al., 2014*; *Kang et al., 2018b*; *Chang et al., 2016*; *Ekanayaka & WM, 2016*). The combination of TGF-$\beta$ ligands and receptors on the cell membrane activates the TGF-$\beta$ signalling and then phosphorylates Smad protein2/3 (smad2/3). The phosphorylated smad2/3 associate with acetylated smad4 becoming Smad2/3/4 complex which translocates into nucleus to recognize EMT-associated transcription factor (EMT-TFs) to initiate gene transcription and proceed with the EMT program (*Huang et al., 2022*; *Chang et al., 2016*; *Fuxe, Vincent & de Garcia Herreros, 2010*). At the same time Smad2/3/4 complex binds to co-activator CBP/p300, and promotes TGF-$\beta$-regulated cancer progression (*Mirzaei & Faghihloo, 2018*). At the nucleus, SIRT1 attaches to the promoter region of TGF-$\beta$, inhibits CBP/p300-mediated acetylation and leads to transcriptional suppression of TGF-$\beta$-mediated oral cancer progression (*Islam et al., 2019*). SIRT1 inhibits the EMT process in oral cancer by inhibiting phosphorylation of smad2/3 and deacetylating Smad4. This inhibits the formation of the SMAD complex thereby repressing the effects of TGF-$\beta$ signalling (*Chen et al., 2014*). On the other hand, SIRT1 hypermethylation has been linked to oral carcinogenesis. SIRT1 is significantly hypermethylated in OSCC tissue samples from betel quid chewers and nonchewers compared with oral mucosa samples from healthy control subjects. Therefore, SIRT1 hypermethylation can be considered a possible predictive biomarker of malignant transformation in betel quid chewers (*Islam et al., 2020*). Additionally, SIRT1 induces chemoresistance. Studies have shown that SIRT1 overexpression regulates and interferes with chemotherapy and enhances chemoresistance in various cancer cells. SIRT1 prevents cisplatin-induced ROS accumulation in an OSCC cell line (Tca8113) and mediates cisplatin resistance (*Xiong et al., 2011*). SIRT1 reportedly promotes autophagy by deacetylating multiple autophagy-related genes (*Sun et al., 2015*). However, one study showed that capsaicin inhibits SIRT1 to enhance the acetylation of

unc-51-like autophagy activating kinase 1 (ULK1) to trigger autophagy in oral cancer cells (*Chang et al., 2020*), which suggests that SIRT1 may inhibit autophagy in oral cancer cells.

## SIRT3

Similar to SIRT1, SIRT3 may function as either an oncogene or suppressor in oral cancer. One study showed that SIRT3 is overexpressed in three OSCC cell lines (HSC-3, UM-SCC-1, and UMSCC-17B) and in OSCC tissues. Downregulation of SIRT3 inhibits OSCC cell growth and proliferation and increases OSCC cell sensitivity to radiation and cisplatin treatments *in vitro* (*Alhazzazi et al., 2011*; *Alhazzazi et al., 2016*). This finding suggests a role for SIRT3 in promoting the development of oral cancer (*Alhazzazi et al., 2011*). SIRT3 is localized in the mitochondria and plays an important role in maintaining the mitochondrial redox balance. Down-regulation of SIRT3 inhibits cell growth and proliferation and promoted apoptosis in OSCC by increasing ROS levels in mitochondria and increasing mitochondrial proteins such as NDUFA9 and GDH acetylation then causing mitochondrial fission (*Alhazzazi et al., 2016*). On the other hand, some studies have shown that SIRT3 may act as an inhibitor of oral cancer cells. In two OSCC cell lines (HSC-3 and OECM1), SIRT3 expression is slightly higher than in normal primary human oral keratinocytes (HOK cells). Surprisingly, it was found that the levels of SIRT3 deacetylase activity in OSCC cell lines were markedly lower than those in HOK cells. Specifically, a mutation closer to the SIRT3 protein's active site reduces the overall enzymatic efficiency of deacetylation, thereby reducing the growth of OSCC cells as a result (*Chen et al., 2013a*). MicroRNA miR-31 is an oncogenic factor in OSCC. There is evidence to suggest that SIRT3 expression reduces miR-31-dependent tumour invasion and migration. It has been shown that miR-31 alteration can decrease mitochondrial membrane potential (MMP), disrupt mitochondrial structure and function by increasing ROS levels, and modulate metabolic switch in OSCC cells (*Kao et al., 2019*). Both of the above-mentioned studies indicate that SIRT3 may play a role in tumor suppression in OSCC.

## SIRT7

SIRT7 functions as a tumour suppressor by suppressing cell proliferation, migration, and invasiveness. One study found that SIRT7 is significantly downregulated in OSCC cell lines and human OSCC tissues with lymph node metastasis. These findings suggest that SIRT7 suppresses EMT in OSCC metastasis by promoting SMAD4 deacetylation (*Li, Zhu & Qin, 2018*). Researchers have further found that miR-770 is an upstream regulator of SIRT7 and that miR-770 promotes OSCC cell migration and invasion through SIRT7/Smad4 signalling (*Jia et al., 2021*). However, another study showed that SIRT7 expression levels do not differ significantly in OSCC tissues, even though SIRT7 is overexpressed in three OSCC cell lines (HSC-3, UM-SCC-1, and UMSCC-17B) compared with primary keratinocytes (*Alhazzazi et al., 2011*).

## Other SIRTs

Currently, the relationship between SIRT2, SIRT4, and SIRT5 and oral cancer have not been investigated. A study showed that the expression of SIRT6 is upregulated in oral SCC samples, which implies that SIRT6 might be associated with SCC development (*Lefort et al.,*

**Table 2  A summary of laboratory evidence of sirtuins and oral carcinogenesis.**

| Sirtuins | Molecular targets and regulatory processes in oral carcinogenesis |
|---|---|
| SIRT1 | **SIRT1 acts as a potential tumor suppressor** |
| | 1. Sirtuin1 inhibits the EMT process in oral cancer by (1) inhibiting the phosphorylation of Smad2/3 and deacetylating Smad4 to suppress the nuclear translocation of Complex Smad2/3/4, (2) repressing the effect of TGF-β signalling on matrix metalloproteinase-7 (MMP7), (3) suppressing CBP/p300-mediated acetylation by binding of TGF-β promoter region, (4) upregulating the expression of epithelial marker E-cadherin and suppressing the expression of myogenic markers, N-cadherin and vimentin. |
| | 2. SIRT1 is involved in downregulating the expression of genes related to migration and invasion, including *csk2a2*, *fra1*, *actb*, and *slug*. |
| | **SIRT1 acts as a potential tumor promoter** |
| | 1. SIRT1-mediated autophagy through ULK1 facilitates the resistance of oral cancer cells from chemotherapy. |
| | 2. Sirt1 mediates cisplatin resistance by preventing cisplatin-induced ROS accumulation in OSCC cell lines. ROS decline promotes proliferation, migration, and invasion of cancer cells. |
| SIRT3 | **SIRT3 acts as a potential tumor suppressor** |
| | 1. A mutation of SIRT3 protein reduces the overall enzymatic efficiency of deacetylation, which leads to the inhibition of cell growth in two OSCC cell lines, HSC-3 and OECM1. |
| | 2. SIRT3 reduces miR-31-dependent tumour invasion and migration by decreasing mitochondrial membrane potential and disrupt mitochondrial structure and function in OSCC cells. |
| | **SIRT3 acts as a potential tumor promoter** |
| | 1. Downregulation of SIRT3 increases OSCC cell sensitivity to radiation and chemotherapy. |
| | 2. Downregulation of SIRT3 causes mitochondrial damage through ROS-induced MMP reduction, which inhibits the growth and proliferation of cancer cells. |
| | 3. Downregulation of SIRT3 promotes apoptosis in OSCC by acetylating mitochondrial proteins NDUFA9 and GDH. |
| SIRT7 | SIRT7 suppresses EMT by promoting Smad4 deacetylation, which results in a decrease in cell proliferation, migration, and invasion. |

**Notes.**

EMT, Epithelial–mesenchymal transition; GDH, Glutamate dehydrogenase; MMP7, Matrix metalloproteinase 7; OSCC, Oral squamous cell carcinoma; ROS, Reactive oxygen species; TGF-β, Transforming growth factor-beta; ULK1, Unc-51-like autophagy activating kinase 1.

*2013*). But another study demonstrated that the expression levels of SIRT1, SIRT2, SIRT3, SIRT5, SIRT6, and SIRT7 were significantly downregulated in cancerous tissues compared with noncancerous tissues (*Lai et al., 2013*). Therefore, more studies are warranted to confirm the role of SIRTs in oral cancer.

### Sirtuins in the treatment of oral cancer

Activators and inhibitors of sirtuins have been developed in recent years, and to date, some activators may be promising drugs in the treatment of oral cancer. For instance, curcumin-induced apoptosis in HNSCC cell lines (FaDu and Cal27 cells) is associated with activation of the SIRT1 signalling pathway. Increasing SIRT1 through curcumin has shown beneficial effects in a xenograft mouse model. These results indicate that SIRT1 may represent an attractive therapeutic target (*Hu et al., 2015*). CAY1059, another SIRT1 activator, suppresses cell growth and migration activity in gingival squamous cell carcinoma Ca9-22 cells (*Murofushi et al., 2017b*). A study conducted by Ling Tao also showed that the green tea catechin (–)-epigallocatechin-3-gallate (EGCG) appears to be a promising medicine because it inhibits SIRT3 activity in oral cancer cells but activates SIRT3 in normal cells (*Tao, Park & Lambert, 2015*). LC-0296, a novel SIRT3 inhibitor, can inhibit cell survival and promote apoptosis by increasing ROS levels in head and neck squamous cell carcinoma (HNSCC) cells (*Alhazzazi et al., 2016*). In addition, it has been reported that sirtuins can indicate the prognosis of oral cancer. According to the study, 79.6% of HNSCC samples showed both nuclear and cytoplasmic SIRT1 positivity, and that was associated with good prognosis compared with SIRT-1 negative cases (*Noguchi et al., 2013*). One study, however, found that there was no significant relationship between the expression of SIRT1 and the prognosis of oral cancer (*Seyedmajidi et al., 2019*). There is evidence to suggest that SIRT6 plays a role in tumor homeostasis, which contributes to a poor prognosis in OSCC patients (*Yoshii et al., 2022*). There is also a study that has shown that the expression levels of SIRT6 and SIRT7 are significantly higher in peripheral blood leukocytes of HNSCC patients compared with healthy individuals, and that the levels of SIRT6 and SIRT7 are recovered in patients after surgery (*Lu et al., 2014*). Based on these results, it may be possible that SIRT6 and SIRT7 are not only potential circulating prognostic markers for HNSCC, but also novel targets for the treatment of this cancer.

## TARGETING SIRTUINS AS A THERAPEUTIC STRATEGY IN CLINICAL TRIALS

Multiple clinical trials of targeting sirtuins, including activators and inhibitors of sirtuins such as resveratrol, quercetin, melatonin, and berberine, in metabolic diseases are in progress (Table 3).

Resveratrol, one of the most extensively studied SIRT1 activators, has been studied for its potential to treat type 2 diabetes(T2D) (*Ma et al., 2016*). One clinical trial (NCT01677611) focused on the effects of resveratrol on skeletal muscle SIRT1 expression in adults with T2D ($n = 10$, 500 mg per day to 3 g per day in three divided doses for a total of 3 months). Quercetin can alleviate insulin resistance and improve glucose metabolism by increasing SIRT1 expression (*Hu et al., 2020*). A completed clinical study of the use of quercetin for

**Table 3** A summary of clinical trials with targeting sirtuins: focusing on metabolic diseases.

| Conditions | Targeting Sirtuins | Intervention | Main endpoints | Phase | NCT or References |
|---|---|---|---|---|---|
| | Resveratrol | 500 mg on Day 1 and increased by 500 mg per day every 3 days to a maximum dose of 3 g per day in three divided for 3 months | Skeletal muscle SIRT1 expression | Phase 1 | NCT01677611 |
| Type 2 Diabetes | Quercetin | 250 mg; oral single dose of 2,000 mg | Glucose tolerance following a maltose tolerance test | Phase 2 | NCT01839344 |
| | Melatonin | 3 mg once daily | Fasting blood sugar; HbA1c | Early Phase 1 | NCT02691897 |
| Type 2 Diabetes Dyslipidaemia | Berberine | 1.0 g daily for 3 months | Glucose levels; HbA1c; HDL-c; LDL-c; Serum triglycerides; Total cholesterol | Phase 3 | NCT00462046 |
| SIRT3 | Curcumin | 80 mg tid, for 6 weeks | C-reactive protein | Phase 2 | NCT01925547 |
| | Melatonin | 8 mg one hour before bedtime for 10 weeks | Metabolic syndrome components | Phase 2 | NCT01038921 |
| Nonalcoholic Fatty Liver Disease | Berberine | 0.5 g tid, for 16 weeks | Improved metabolic parameters | Phase 2 | NCT00633282 |

patients with diabetes also showed that 2,000 mg of oral quercetin resulted in a decrease in postprandial blood glucose levels (NCT01839344). SIRT1 and SIRT3 are key melatonin targets; in rats, melatonin efficiently alleviates glucose metabolism disorders by decreasing mitochondrial dysfunction through activating SIRT1 and SIRT3 (*Chen et al., 2019*; *Zhang et al., 2017*). Several decades of clinical studies of melatonin for a variety of diseases have been performed. A phase 2 clinical trial of melatonin in 39 metabolic syndrome patients (8 mg of melatonin) for 10 weeks showed that melatonin was relatively safe and improved at least one of the five components associated with metabolic syndrome. A recent study of 3 mg melatonin per day for 3 months in 60 participants with T2D (aged from 20 years to 65 years) assessed the efficacy of melatonin in the control of blood sugar (NCT02691897). Curcumin can inhibit apoptosis *via* Sirt1-Foxo1 signalling in rats with type 2 diabetes (*Ren et al., 2020*) and enhances lipid metabolism in adipocytes by promoting AMPK activity by activating SIRT1 (*Ejaz et al., 2009*). Curcumin has potential value in preventing metabolic syndrome. A clinical study investigated the effect of curcumin on inflammation and lipid metabolism markers in subjects at risk for metabolic syndrome, and the data reported that the consumption of 98 mg of highly bioavailable curcuminoids was safe with slightly elevated blood cholesterol and C-reactive protein levels (NCT01925547). Berberine mediates glucose and lipid metabolism *via* SIRT1 signalling (*Pang et al., 2015*; *Hasanein, Ghafari-Vahed & Khodadadi, 2017*). Clinical trials with berberine in relation to metabolic syndrome have been reported. One trial was focused on the efficacy and safety of berberine in the treatment of 116 T2D patients with dyslipidaemia (1.0 g daily for 3 months), with the results showing berberine to be effective and safe in the treatment of persons with

diabetes and dyslipidaemia (NCT00462046). In another clinical trial, berberine, as a new cholesterol-lowering drug, was effective for alleviating non-alcoholic fatty liver disease by improving lipid metabolism (NCT00633282). Studies targeting sirtuins in relation to T2D, dyslipidaemias, metabolic syndrome, and non-alcoholic fatty liver disease are being conducted or recruiting participants, and results from these clinical trials will likely reveal the potential of sirtuins in humans. Although sirtuins play a significant role in regulating oral cancers, there are few clinical trials with activators or inhibitors of sirtuins related to oral cancers and many unanswered questions surrounding sirtuin-regulated oral cancers that need to be further addressed.

## CONCLUDING REMARKS

The past decade has shown notable progress towards an understanding of the role of sirtuins in metabolic regulation and tumorigenesis. This is particularly relevant in oral cancer, which can be viewed as both a metabolic and genetic disorder. The role of sirtuins in the carcinogenesis of oral cancer is still unclear due to the limited number of studies and contradictory findings. However, a deeper understanding of SIRT biology at both the molecular and physiological levels will be critical in order to determine the potential therapeutic benefits of activating (SIRT activators) or inactivating SIRTs (SIRT inhibitors) and their detrimental side effects.

## ACKNOWLEDGEMENTS

In this article, the authors acknowledge the many investigators whose publications they were unable to cite due to space restrictions. Figures were generated using the subscription software BioRender (Toronto, ON, Canada).

### Funding

This review was supported by grants from the National Natural Science Foundation of China (NO. 81700967) and the Natural Science Foundation of Hunan Province (NO. 2018JJ3706). He-Ling Wang was also funded by the China Scholarship Council, start date was 19 January 2020. The funders had no role in study design, data collection and analysis, decision to publish, or preparation of the manuscript.

### Competing Interests

The authors declare there are no competing interests.

### Author Contributions

- Xu Quan conceived and designed the experiments, performed the experiments, analyzed the data, prepared figures and/or tables, authored or reviewed drafts of the article, and approved the final draft.
- Ying Xin conceived and designed the experiments, performed the experiments, analyzed the data, authored or reviewed drafts of the article, and approved the final draft.
- He-Ling Wang conceived and designed the experiments, performed the experiments, analyzed the data, prepared figures and/or tables, authored or reviewed drafts of the article, and approved the final draft.
- Yingjie Sun analyzed the data, prepared figures and/or tables, authored or reviewed drafts of the article, and approved the final draft.
- Chanchan Chen performed the experiments, analyzed the data, authored or reviewed drafts of the article, and approved the final draft.
- Jiangying Zhang conceived and designed the experiments, analyzed the data, prepared figures and/or tables, authored or reviewed drafts of the article, and approved the final draft.

## Data Availability

This literature review did not include raw data.

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
