# Peer review of "Implications of altered sirtuins in metabolic regulation and oral cancer"

_PeerJ, doi:10.7717/peerj.14752_

## Round 0.1 · original submission · Major Revisions

Reviewers have raised some concerns in the form of MAJOR revision, which requires substantial and thorough revision to appreciate the quality of the manuscript for publication in PeerJ. Therefore, authors are requested to revise their manuscript in light of reviewers comments and resubmit accordingly.

Reviewer 1 ·

Basic reporting

.

Experimental design

.

Validity of the findings

.

Additional comments

MAJOR REVISIONS
-The authors have presented an interesting review entitled Implications of altered sirtuins in metabolic regulation and oral cancer. This present manuscript builds on understand SIRT biology at both the molecular and physiological levels.
-I suggest adding a figure or table that summarizes the most representative targets of SIRTS and their metabolic regulation.
-This review also emphasizes about the possible relationships of sirtuins with oral cancers and novel insight into new therapeutic approaches targeting sirtuins that may lead to eûective strategies for combating oral malignancy. However, I suggest you add a new section with the possible therapeutic of SIRTs (using inhibitors o activators of SIRTs) approach in oral cancer to expand upon the knowledge gap being filled.
-I suggest you a schematic overview of regulators and molecular targets of SIRT that could be associated in oral carcinogenesis or a schematic overview of most relevant sirtuin inhibitors and activators.

·

Basic reporting

The manuscript entitled "Implications of altered sirtuins in metabolic regulation and
oral cancer" is review that addresses the all the sirtuins identified in humans, along with main characteristics, activity domains, main modifying activity along with others reported. However, though it mentions that sirtuins are also involved in many activities, metabolism, DNA repair among them, this is not described, only mentioned, much less its involvement in oral cancer, thus I would suggest to include some of this in this complete review.

Experimental design

No comments as this is a review, not a regular scientific paper

Validity of the findings

Same as above, no comments

Additional comments

The review entitled: "Implications of altered sirtuins in metabolic regulation and
oral cancer is well described, its reading is easy to follow, the authors did a good job in covering everything about sirtuins, however, DNA repair is not described at all, and though the papers analyzed and cited do not state any relation to DNA repair, this cannot be discarded.

·

Basic reporting

The topic of the review has a cross-disciplinary interest, including clinicians (particularly specialist in oral medicine) and researchers in the field of cancer and metabolism.
Notwithstanding, the role of sirtuins in head and neck squamous cell carcinoma has been recently reviewed by Ezhilarasan et al. and few experimental studies have been published afterwards.
In the introduction, the authors could provide some background on oral cancer epidemiology, pathology (including cellular and molecular mechanisms of tumorigenesis) and medical treatments, to meet the audience spectrum of clinicians and researchers that may not be familiarized with this topic.

Experimental design

The most important issue is the focus of this review.
The authors claim to provide an overview and an update on sirtuin functions in metabolism and with a specific focus on their role in oral cancer. However, the role of sirtuins in general metabolism and its implications in metabolic diseases (diabetes, nonalcoholic fatty liver disease…) deserved great attention (an all section), while the specific role in oral cancer was poorly explored.
Perhaps the authors could briefly explain the classification, structure and subcellular localization of sirtuins, condense the information regarding the role of sirtuins in the general metabolism and discuss in more detail the regulatory mechanisms (metabolic and others) employed by sirtuins to modulate oral cancer.
The same commentary about the figures, which are not related to OSCC carcinogenesis nor the metabolic functions of sirtuins, the topics intended to be covered by authors. Besides, readers can find several other published articles with a focus on the structure, classification and subcellular localization of sirtuins. Instead of the figures included, authors could elaborate a schematic figure with the potential regulatory mechanisms of sirtuins in carcinogenesis.
It would also be valuable to summarize the outcomes of pre-clinical and clinical studies in a table.


Another important issue is the findings to support what the authors state in their conclusions: “The past decade has shown notable progress toward an understanding of the role of sirtuins in metabolic regulation and tumorigenesis. This is particularly relevant in oral cancer, which can be viewed as both a metabolic and genetic disorder, in that the metabolic pathways involved in oral cancer and potentially regulated by sirtuin are not properly explored. For instance, the inhibitor of SIRT3, LC-0296, promoted apoptosis and inhibited cell survival and proliferation of HNSCC cell lines. This can be explained by the role of SIRT3 in maintaining the mitochondrial redox balance since LC-0296 treatments caused significant upregulation of intracellular ROS in HNSCC cell lines.


Other issues:
L42. The authors state there is no comprehensive review on the implication of altered sirtuin in oral cancer. However, Ezhilarasan et al. (reference number 134) have recently published a review on this subject.
L444. Can the authors explain what is meant by “SIRT1 acts as a bifunctional in oral cancer.”? Have the authors found evidence on the role of SIRT1 in metastatic ability of OSCC cells?
L477. “This study showed…” – to which study are the authors referring to? The reference cited (24) doesn’t seem to be related.
Evidence from other cancer cells could be added to support the potential molecular and cellular implications of sirtuins in oral cancer.
Authors could include the findings of other studies, namely:
 Alhazzazi et al., 2016. A novel sirtuin-3 inhibitor, LC-0296, inhibits cell survival and proliferation, and promotes apoptosis of head and neck cancer cells.
 Li et al., 2018. Epithelial mesenchymal transition induced by the CXCL9/CXCR3 axis through AKT activation promotes invasion and metastasis in tongue squamous cell carcinoma.
 Ezhilarasan et al., 2019. Syzygium cumini extract induced reactive oxygen species-mediated
 Apoptosis in human oral squamous carcinoma cells.
 Hu et al., 2015. Curcumin as therapeutics for the treatment of head and neck squamous cell carcinoma by activating SIRT1.
 Lu et al., 2014. The potential of SIRT6 and SIRT7 as circulating markers for head and neck squamous cell carcinoma.
 Mishev et a., 2014. Prognostic value of matrix metalloproteinases in oral squamous cell carcinoma.
 Noguchi et al., 2013. SIRT1 expression is associated with good prognosis for head and neck squamous cell carcinoma patients.
 Seyedmajidi et al., 2019. Immunohistochemical expression of SIRT1 in oral squamous cell carcinoma and its relationship with clinical-pathological factors.

Validity of the findings

In their conclusions, authors should also address that the role of sirtuins in the carcinogenesis of oral cancer is still unclear due to the limited number of studies and contradictory findings, Authors could specify potential targets that need to be explored with SIRT inhibitors and activators in future studies.

---

## Round 0.2 · Major Revisions

Though the manuscript is significantly improved by the authors, reviewer 3 still has raised some concerns and provided suggestions to improve the manuscript. Please revise considering the comments and resubmit.

·

Basic reporting

The manuscript entitled: Implications of altered sirtuins in metabolic
regulation and oral cancer has been revised by authors and a substantial improvement is noticed.
Thus I recommend its publication

Experimental design

It does not apply, it is a review.

Validity of the findings

Same

Additional comments

None

·

Basic reporting

The authors revised the text attending to most of the issues pointed out by the reviewer, clearly improving and enriching their review. The disperse information regarding the regulatory factors of sirtuins implicated in metabolism and oral carcinogenesis are now displayed and summarized in tables.
Nevertheless, due to the complex role of sirtuins (and contradictory findings) in oral carcinogenesis, it is still difficult to assess the information and to understand the molecular targets of each sirtuin. Therefore, as suggested by another reviewer, it would be valuable to provide a schematic overview with the molecular targets of sirtuins implicated in oral carcinogenesis.

Experimental design

.

Validity of the findings

.

---

## Round 0.3 · accepted · Accept

Manuscript is significantly improved by the authors and now can be accepted in its current form.